# MoS_2_@ZnO Nanoheterostructures Prepared by Electrospark Erosion for Photocatalytic Applications

**DOI:** 10.3390/nano11010157

**Published:** 2021-01-09

**Authors:** Vladimir An, Herman Potgieter, Natalia Usoltseva, Damir Valiev, Sergei Stepanov, Alexey Pustovalov, Arsenii Baryshnikov, Maksim Titov, Alesya Dolinina

**Affiliations:** 1Kizhner Research Center, School of Advanced Manufacturing Technologies, Tomsk Polytechnic University, Lenin avenue, 30, 634050 Tomsk, Russia; usolceva@tpu.ru (N.U.); rts.gsd1997@gmail.com (A.B.); titov081197@gmail.com (M.T.); asa87@tpu.ru (A.D.); 2Department of Natural Sciences, Manchester Metropolitan University, Manchester M15 6GD, UK; h.potgeiter@mmu.ac.uk; 3Division of Materials Science, School of Advanced Manufacturing Technologies, Tomsk Polytechnic University, Lenin avenue, 30, 634050 Tomsk, Russia; rubinfc@tpu.ru (D.V.); stepanovsa@tpu.ru (S.S.); 4R&D Laboratory for Clean Water, School of Advanced Manufacturing Technologies, Tomsk Polytechnic University, Lenin avenue, 30, 634050 Tomsk, Russia; pustovalov@tpu.ru

**Keywords:** electrospark erosion, heterostructures, photocatalytic water splitting, MoS_2_@ZnO, electrical explosion of wires, self-propagating high-temperature synthesis (SHS)

## Abstract

MoS_2_@ZnO nanoheterostructures were synthesized by electrospark erosion of zinc granules in a hydrogen peroxide solution and simultaneous addition of MoS_2_ nanostructured powder into the reaction zone. The morphology, size of the crystallites, as well as elemental and phase composition of the prepared structures, were examined using transmission electron microscopy and X-ray diffraction analysis. It was found that the synthesized products represent heterostructures containing MoS_2_ nanoparticles formed on ZnO nanoparticles. Raman spectroscopy and photoluminescence analysis were also used for characterization of the prepared heterostructures. The obtained MoS_2_@ZnO nanostructures revealed an intense broad emission band ranging from 425 to 625 nm for samples with different fractions of MoS_2_. Photocatalytic measurements showed that the maximal hydrogen evolution rate of the prepared nanoheterostructures was about 906.6 μmol·g^−1^·h^−1^. The potential of their application in photocatalytic water splitting was also estimated.

## 1. Introduction

During recent decades, functional heterostructured metal oxide and sulfide materials received much interest from scientists working in the field of materials for photocatalytic applications [1,2,3,4]. Non-equilibrium fabrication methods like self-propagating high-temperature and mechanochemical synthesis, electrical explosion of wires, and electrospark erosion methods have displayed tremendous potential for synthesizing such materials [5,6,7]. These fabrication methods of such materials yielded structures that revealed high reactivity in various processes. Solar energy is a well-known alternative energy source, and photocatalytic water splitting using sunlight is a promising and eco-friendly approach to produce hydrogen. Semiconductor materials can be used for water splitting if the lowest level of the conduction band (CB) is more negative than the H^+^/H_2_ reduction potential (0 V vs. the normal hydrogen electrode (NHE)), and the highest level of valence band is more positive than the H_2_O/O_2_ oxidation potential (1.23 V vs. NHE). [8] Combinations of n-type metal oxides and p-type transition metal dichalcogenides (TMDs) result in a semiconductor n-p heterojunction which can absorb a wide spectrum of solar energy containing both UV and visible light [9,10,11].

Among the materials having a high potential for photocatalytic water splitting, MoS_2_ is one of the most interesting, because of its thermal and optoelectronic performance [12,13]. Theoretical predictions demonstrate that MoS_2_ has a high potential as a hydrogen evolution reaction (HER) catalyst, because of its low Gibbs free energy for hydrogen adsorption ΔG_H_, which is close to that of noble metals [14]. For photocatalytic reactions mediated by semiconductors, ZnO is also a very good candidate since it possesses high stability, a wide bandgap and non-toxic properties [15]. At the same time the high recombination rate of charge carriers is the reason for limited photocatalytic ZnO activity. MoS_2_ is active at visible light wavelengths, whereas ZnO is excited by ultraviolet (UV) light. The combination of ZnO and MoS_2_ can reduce the band gap, increase the light absorption range and enhance the recombination of photogenerated electron-hole pairs of ZnO, and thus improve the photocatalytic ability of ZnO in the visible light region [16,17,18].

Bulk or multilayer MoS_2_ and ZnO are semiconductors with indirect bandgaps, while bandgaps of the single layered ones are direct. ZnO/MoS_2_ heterostructures reported [19] to cause band alignment of the ZnO/MoS_2_ interface. Type-II band alignment features spatial separation of the valence band (VB) and the conduction band (CB) [4,17,20].

The photogenerated electron-hole pairs in MoS_2_ are suitable for the spatial zones of the water-splitting reaction, which provide a high rate of recombination. Theoretical calculations reported in [4] have shown that a MoS_2_-based heterostructure formed by stacking MoS_2_ on two-dimensional zinc oxide (ZnO) can potentially be an excellent composite for photocatalytic applications. It has been shown that the energy levels of both water oxidation and reduction lie within the bandgap of the MoS_2_/ZnO heterostructure, which provide for efficient water splitting. At the same time, a closer MoS_2_-ZnO junction can accelerate new active sites and impact the recombination of the electron–hole pairs, which provides an increase in the photon utilization rate [21]. The contribution of active species such as superoxide ($O_{2}^{-}$), hydroxyl radical ($OH^{-}$), and hole ($h^{+}$) in the photocatalytic process is discussed in [17].

Hydrothermal synthesis in an autoclave has been widely used to prepare MoS_2_@ZnO heterostructures. Different ways of MoS_2_ hydrothermal synthesis which were developed include: precipitation from solutions of molybdenum and sulfur salts [22], or the solution of molybdenum salt and sulfur [18]. Grinding of bulk MoS_2_ can result in nanosheets [22]. MoS_2_ synthesis, followed by ZnO preparation in a zinc salt solution including the dispersed MoS_2_ powder; ultrasonication [22] or vigorous stirring [18,23,24] contribute to the MoS_2_ homogenization and preparation of a MoS_2_/ZnO heterostructure.

It is proposed [23] to illuminate a zinc salt solution with MoS_2_ powder with a near-infrared (NIR) laser to drive the growth of ZnO nanoparticles on the surface of the MoS_2_ by heat energy from the MoS2 nanosheets photothermal conversation.

ZnO hydrothermal synthesis can be achieved by growing ZnO nanorods in a zinc salt solution [25]. Magnetron sputtering of Zn targets also produces ZnO nanorods [25]. A MoS_2_/ZnO heterostructure is formed by adding ZnO nanorods to an autoclave filled with molybdenum and sulfur salts [26,27]. Another way to prepare a MoS_2_/ZnO heterostructure is by adding ZnO powder to a MoS_2_ suspension [16,28]. MoS_2_/ZnO heterostructures can be doped to improve their photocatalytic properties [29].

Along with dispersed heterostructures, coated heterostructures are used as photocatalyst for the hydrogen evolution reaction. Substrates which can be utilize include fluorine doped tin oxide (FTO) glass [30], reduced graphene oxide [8], and single crystal silicon coated with a thin layer of Au [31].

Some kind of hydrothermal condition occurs during the electrospark erosion of metals. This results in preparation of metal, oxide and carbides nanoparticles under non-equilibrium conditions in different liquid media [7,32]. Adding as-prepared metal compounds to liquid medium contributes the heterostructure formation.

A photoluminescence (PL) investigation is a convenient approach to disclose the energy band structure of materials (electron-hole generation and recombination) [33]. The light absorption spectra of MoS_2_, ZnO and the MoS_2_/ZnO heterostructure depend upon the synthesis conditions [34,35]. A schematic diagram of charge generation and transfer process under ultraviolet (UV) and visible (Vis) excitation is proposed to understand the photoluminescence mechanism [34]. The bandgap of MoS_2_/ZnO heterostructures can be adjusted by changing the MoS_2_/ZnO ratio. Photoluminescence spectra have revealed that there is the red shift in the band-edge emission wavelength for the MoS_2_/ZnO heterostructure compared with ZnO, because of a decrease of bandgap [34]. When ZnO is added to MoS_2_, it causes a blue shift due to an increase of the bandgap [27].

The photocatalytic activity of heterostructures depends on the ZnO morphology and MoS_2_ content. The preparation method influences the content at which the photocatalytic activity reaches a peak [24,27,36,37]. Nanorods provide the best photocatalytic activity of the ZnO heterostructure [38].

Photocatalytic activity can be adjusted by adding Na_2_S and Na_2_SO_3_ to an aqueous solution. In situ formed ZnS slows down the recombination of electron-hole pairs, and therefore increases the photocatalytic H_2_ evolution rate [8,37].

The combination of hydrothermal methods with other non-equilibrium fabrication methods listed above creates an additional opportunity to prepare new heterostructures with enhanced characteristics and operating performance (e.g., phase composition, donor-acceptor pair recombination, the rate of photocatalytic activity in water splitting reactions). Compared with the methods described, the combination of electrical explosion of molybdenum wires [5], self-propagating high-temperature synthesis of molybdenum sulfide [39] and electrospark erosion of zinc [7,32] provides a technique for preparing heterostructures from simple elements (Mo, S, Zn). It makes the synthesis more ecologically friendly because reduces the amount of waste water.

This work is aimed at fabrication and characterization of MoS_2_@ZnO heterostructures for photocatalytic applications by electrospark erosion (ESE) of zinc granules in a hydrogen peroxide solution and simultaneous addition of MoS_2_ nanostructured powder in the reaction zone. MoS_2_ as-prepared by self-propagating high-temperature synthesis was added to solution. The main aim was to use the non-equilibrium conditions of the ESE method where newly formed zinc oxide nanoparticles adhered to molybdenum disulfide nanolamellar particles which are separated by the electrosparks. Photoluminescent and hydrogen evolution measurements will be carried out to reveal the application of heterostructures in photocatalytic processes.

## 2. Materials and Methods

### 2.1. Fabrication of Nanostructured MoS_2_ by Self-Propagating High-Temperature Synthesis

Nanostructured MoS_2_ was synthesized by a self-propagating high-temperature reaction using mixtures of molybdenum nanopowder and pure elemental sulfur (99.90%, ZAO “Soyuzhimprom”, Novosibirsk, Russia) [39]. Molybdenum nanopowder was prepared using the method of electrical explosion of wires in argon, described in [6]. A 0.25 mm diameter Mo wire (99.96%, OOO “GK “SMM”, Moscow, Russia) was used. The following parameters were applied in the electrical explosion: voltage—28 kV, the exploded wire distance between the electrodes—100 mm, argon pressure—1 atm. According to BET measurements, the surface area A_S_ and surface-average particle size D_S_ of the powders were 4.6 m^2^·g^−1^ and 130 nm, respectively.

In the synthesis of the nanostructured MoS_2_, stoichiometric mixtures of electroexplosive molybdenum nanopowder and pure elemental sulfur were used. These were then compacted into cylindrical pellets of 30 mm diameter and 10 mm height and placed into the sample holder in a special high pressure reactor described in [39]. The process of self-propagating synthesis was initiated by point-heating of the pellet top using a Nichrome wire connected to a direct current (DC) power supply. Once started, a wave of exothermic self-propagating reaction sweeps through the remaining pellet material. The process temperature was controlled by a W/W-Re thermocouple connected to an oscilloscope employed for recording the thermo-emf signal. After completion of the self-propagating high-temperature synthesis (SHS) reaction, the final products were cooled in the reaction chamber. The SHS products represented silver gray packets of nanostructured molybdenum disulfide, according to X-ray diffraction phase analysis conducted.

### 2.2. Synthesis of MoS_2_-ZnO Heterostructures

The electrospark erosion method was used for the synthesis of the MoS_2_@ZnO nanoheterostructures [7]. In our case, a 38% aqueous solution of hydrogen peroxide was employed. MoS_2_@ZnO nanoheterostructures were synthesized by electrospark erosion of zinc granules (analytic grade, ZAO “Soyuzhimprom”, Novosibirsk, Russia) in the hydrogen peroxide solution (OOO “INNOVATSIIA”, Voronezh, Russia) and simultaneous addition of MoS_2_ nanostructured powder in the reaction zone. Figure 1 shows the scheme of the electrospark erosion experiment for the fabrication of MoS_2_@ZnO heterostructures. The reactor represents a porcelain vessel very resistant to electric current pulses. 185 g of zinc granules were placed into the reactor and then 200 mL of hydrogen peroxide solution were poured. Two zinc electrodes were immersed in the hydrogen peroxide solution until the complete electric contact with the zinc granules. After this procedure, electric current pulses were applied to the electrodes from the power supply within 30 s for each experiment. The experimental conditions were the following: the electrode separation distance was 10 cm, the voltage was 500 V, the current was 150 A. After the experiment, the prepared suspension was separated into two fractions by decantation. Both fractions were then maintained at 80 °C in a drying oven for 1 h.

### 2.3. Characterization Techniques

X-ray diffraction (XRD) measurements were performed using a Shimadzu-7000S diffractometer (CuKα radiation, *λ* = 1.5418 Å, Shimadzu, Kyoto, Japan). The phase composition and the crystallite size (coherent scattering region) of the prepared structures were determined. Transmission electron microscopy (TEM) analysis was performed on a JEOL JEM-2100F (JEOL Ltd., Tokyo, Japan) in order to examine the morphology of the prepared nanoheterostructures. Raman spectroscopy measurements were conducted using a Centaur I HR spectrometer (OOO “Nano Scan Technology”, Dolgoprudny, Russia) at a wavelength of 538.8 nm. This instrument combines a scanning probe microscope, as well as a confocal microscope/spectrometer with double dispersion for obtaining Raman scattering and fluorescence spectra and spectral images. The diameter of laser spot in Raman scattering measurements was 3 μm. The acquisition time was 10 min per sample. The number of investigated areas was varied from 3 to 5.

The spectral power distribution of the total radiant flux of the PL spectra was measured by using an integrating sphere that was connected to a CCD detector (AvaSpec-3648, (Avantes, Apeldoorn, The Netherlands). Samples were prepared in such a way that the surface area was ~1 cm^2^. The energy efficiency of the phosphors was evaluated as the ratio of the radiation flux of the blue chip and the radiation flux of the phosphors. All measurements were carried out at room temperature. The LED source parameters were 395 nm, 3.25 V, 0.03 A. The hydrogen production rate was measured using a special measurement scheme. 60 mg of the prepared nanostructured powder was dissolved in 100 mL aqueous solution containing 0.1 M Na_2_S and 0.05 Na_2_SO_3_ as the sacrificial reagents. The samples were treated under full spectra irradiation by a 150 W Xe lamp (or deuterium lamp (30 W)) in order to start the photocatalytic reaction. The flask with the sample was connected with a tube to a scaled hermetic burette with which the amount of released hydrogen was measured.

## 3. Results and Discussion

### 3.1. X-ray Phase and Structural Analysis of the Prepared MoS_2_@ZnO Heterostructures

X-ray diffraction measurements were performed in order to determine the phase composition and crystalline properties of the prepared nanoheterostructures, as shown in Figure 2. MoS_2_ revealed diffraction peaks at 14.4°, 29.3°, 38.6°, 44.4°, 47.7° and 54.3°, corresponding to the (002), (004), (101), (103), (006) and (018) hexagonal phases as matched with the standard JCPDS card No. 37-1492. On the other hand, diffraction peaks at 31.2°, 36.3°, 43.1°, 57.7° and 63.2° correspond to the (100), (101), (100), (105) and (110) planes of the hexagonal wurtzite structure of ZnO as matched with the JCPDS card No. 36–1451.

The nanoheterostructure MoS_2_@ZnO shows the main characteristic peaks from both MoS_2_ and ZnO. This can be attributed to the successful formation of nanoheterostructures during the process of electrospark erosion of the zinc granules in the hydrogen peroxide solution with simultaneous addition of the nanostructured molybdenum disulfide. The XRD pattern of MoS_2_ and ZnO also demonstrates the presence of traces of Mo_2_S_3_ and ZnS phases, which were not found in the final products of the SHS and ESE processes. Presumably, they form at the interface between the newly formed ZnO and dispersed MoS_2_ particles. These phases are then condensed on the surface of the ZnO particles as interfacial layers, because of the simultaneous reactions between ZnO and MoS_2_.

### 3.2. Raman Analysis of the Prepared MoS_2_@ZnO Heterostructures

In order to determine the composition of the produced nanoheterostructures, Raman spectroscopy was also used for the characterization of the prepared nanoheterostructures. The Raman spectrum of MoS_2_@ZnO is shown in Figure 3. The synthesized MoS_2_@ZnO showed peaks at 330, 360, 812, 838, 864, 897 and 950 cm^−1^. The Raman peaks at ∼330–360 cm^−1^ are related to molybdenum sulfide vibration modes of ν(Mo–S) [40]. These vibration modes of bridging and terminal S_2_^2−^, as well as vertex sulfide centers are related to the quasi-amorphous state of the MoS*_x_* formed during the electrospark erosion process. The Raman peaks at 950 cm^−1^ was assigned to molybdenum sulfide defects, suggesting the possible presence of molybdenum oxysulfide MoS*_x_*O*_y_* due to the extreme energetical process of electrospark erosion of the zinc granules in the hydrogen peroxide solution. The latter can act as a source of oxygen-containing defects forming in the structure of the nanostructured molybdenum disulfide.

The multi-phonon bands at 775–838 cm^−1^, including the peaks at 812 and 832 cm^−1^, correspond to zinc oxide [41,42]. The anomalous mode detected at 832 cm^−1^ could be due to the influence of MoS_2_ at their interface. Moreover, a special role of the extreme non-equilibrium conditions under which the MoS_2_@ZnO nanoheterostructures were prepared during electrospark erosion in the medium of hydrogen peroxide can be the origin of some of the observed anomalies in their Raman spectrum.

### 3.3. Structural and Compositional Analysis

The morphology of the as-synthesized photocatalytic nanomaterials was studied with transmission electron microscopy (TEM). Figure 4a shows the TEM image of the MoS_2_@ZnO nanoheterostructures prepared using the method of electrospark erosion. It can be seen that the synthesized material represents two different types of nanoscale overlapping particles. The faceted spherical particles of 20–40 nm in size represent wurtzite type hexagonal zinc oxide (ZnO). The black elongated and also hexagonal particles are nanosized sheets of molybdenum disulfide adhering to ZnO nanoparticles.

Energy-dispersive X-ray spectroscopy (EDS) helps to find the purity and elemental composition of the prepared materials. The EDS spectra in Figure 4b show the elemental composition of the prepared MoS_2_@ZnO material. The peaks of Mo, S, Zn and O are clearly visible.

The very small size of the ZnO particles is related to the extreme conditions of the electrospark erosion process [7]. The formation of the ZnO nanoparticles can consist of two simultaneous processes. Firstly, melting of Zn granules and the electrodes when applying voltage pulses with the subsequent release of the material as molten metal droplets with an initial temperature of approximately 2200 K, occurs. Secondly, evaporation of the materials of the medium and the electrodes with plasma formation takes place. Surface interaction of the molten zinc with the hydrogen peroxide aqueous solution occurs during the fast cooling of the particles with a cooling rate of about 10^9^ K·s^−1^. When the process is finished, this interaction results in particles consisting of a metal core coated with a film of interaction products of the metal with the dispersion medium, as well as single particles formed due to radiation-chemical and thermal decomposition of the dispersion medium.

The plasma formation results in atomization of the medium with subsequent recombination and formation of the decomposition products, which then interact with products of the first process to form a highly dispersed composite material. When applying current pulses, a hydromechanical effect accompanied by strong cavitation occurs. Finally, the particles forming are well distributed in the total volume of the suspension. At the same time, the particles start to disperse from the submicron S-Mo-S sandwich packets into nanoplatelets with a thickness of a few layers to form centers onto which newly formed ZnO nanoparticles can adhere. Process behavior under highly non-equilibrium conditions leads to the formation of nanocomposite structures with a high volume and surface deficiency which consequently display a high chemical reactivity in various processes.

### 3.4. Photoluminescent Measurements

The room temperature PL spectra of MoS_2_-coated ZnO nanocomposites are shown in Figure 5a. There is an intense broad emission band ranging from 425 to 625 nm for the samples with different fractions of MoS_2_. The mass of MoS_2_ in the composite MoS_2_@ZnO varies from 1 to 2 g. The broad band at around 503 nm strongly decreases when the MoS_2_ content increased. The reflection also decreased when the MoS_2_ content increased (Figure 5b). The reflection measurement accuracy was ±1%. An obvious quenching effect is observed for defect emission after the ZnO nanoparticle is coated with MoS_2_. The nature of this band is possibly connected with defects emission of the ZnO [43]. The nature of the green emission is contradictory. In [44] it was demonstrated that the green luminescence induced by an oxygen vacancy peaks at 490 nm, while the typical emission of the ZnO nanocomposite located around 503 nm corresponds to a donor-acceptor pair recombination. The cause of the deep level emission in the ZnO is controversial, although it is well accepted that the visible luminescence has multiple centers and pathways [45,46,47]. The emission bands represent a complex composition and could possibly be due to superposition of MoS_2_ and ZnO emissions. We recently have demonstrated the emission of MoS_2_ nanocrystals excited by electrons of nanosecond duration with different energy densities in [48]. With increasing energy densities from 6 up to 200 mJ·cm^−2^ the cathodoluminescence intensity strongly increases in the range of 450–675 nm. The authors suggested that luminescence is attributed to excess electrons in the conduction band. The bandgap of the nanopowder exhibits a high density of states.

The result obtained suggests that the recombination rate of photogenerated carriers is significantly reduced and that the charge transfer between the interface of ZnO and MoS_2_ is evidently facilitated. The reduced emission in the “green” spectral region for the MoS_2_ and ZnO mixture reported in reference [49] supports our assumption.

### 3.5. Hydrogen Evolution Measurements

The photocatalytic H_2_ production activities of the synthesized MoS_2_@ZnO nanoheterostructures were tested in an aqueous solution containing 0.1 M Na_2_S and 0.05 M Na_2_SO_3_ as the sacrificial reagents while applying full spectra irradiation by a 150 W Xe lamp (or deuterium lamp (30 W)). Sodium salts of analytic grade were used to make a solution (Na_2_S·9H_2_O, 97.8%, LC “VEKTON”, Moscow, Russia; Na_2_SO_3_, 98.0%, LC “Karpov chemical plant”, Mendeleyevsk, Russia). A plot of the variation in activities of the MoS_2_–ZnO nanoheterostructures irradiated by the different types of light source (different powers of irradiation) is shown in Figure 6. The sample prepared by electrospark erosion of Zn granules in the peroxide solution with the maximal addition of 2.0 g of MoS_2_ nanolamellar powder exhibits a photocatalytic activity with a H_2_ evolution rate (HER) of 906.6 and 114.9 μmol·g^−1^·h^−1^ when exposed to the irradiation of the xenon or deuterium lamp, respectively. The accuracy of the hydrogen evolution rate measurements is within standard deviation. According to the error analysis for the HER measurements, the standard deviations depend on the amount of MoS_2_ added to the reaction zone and were in the ranges of 8.3–10.7 μmol·g^−1^·h^−1^ and 25.9–30.1 μmol·g^−1^·h^−1^ for the experiments with the use of the deuterium and xenon lamp, respectively. Therefore, it was demonstrated that the hydrogen evolution rate depends on the amount of nanostructured MoS_2_ added into the reaction zone and the lamp irradiation power as a light source as well. These excellent photocatalytic properties ensure a great potential of the prepared nanoheterostructures to be used for efficient photoelectrochemical water splitting.

## 4. Conclusions

The present study demonstrated the successful preparation and synthesis of MoS_2_@ZnO nanoheterostructures by using electrospark erosion of zinc granules in a hydrogen peroxide aqueous solution with the simultaneous addition of nanostructured MoS_2_ powder in the reaction zone. Furthermore, TEM analysis proved that the as-prepared materials have binary heterostructures consisting mostly of MoS_2_ nanoparticles adhered to ZnO. The room temperature PL spectra of MoS_2_-coated ZnO nanoheterostructures showed an intense broad emission band ranging from 425 to 625 nm for the samples with different fractions of MoS_2_. The result obtained suggests that the recombination rate of photogenerated carriers is significantly reduced and that the charge transfer between the interface of ZnO and MoS_2_ is evidently facilitated. Finally, photocatalytic measurements showed that the prepared nanoheterostructures produced an attractively high hydrogen evolution rate of 906.6 μmol·g^−1^·h^−1^ and thus have great potential to be used for efficient photoelectrochemical water splitting.

## Figures and Tables

**Figure 1 nanomaterials-11-00157-f001:**
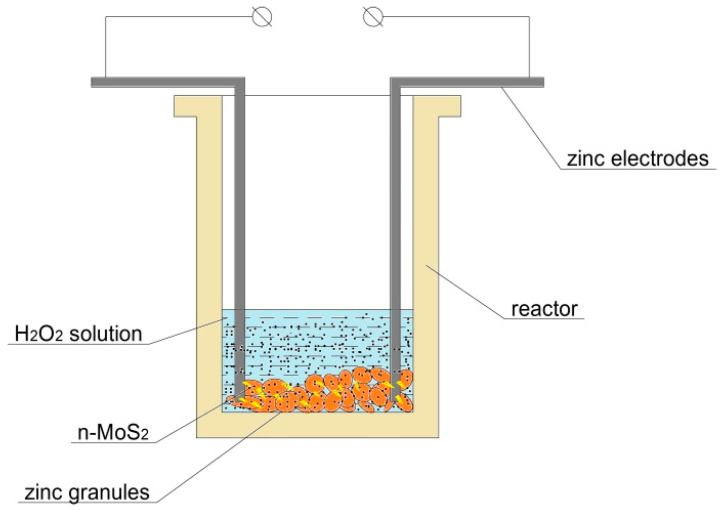
Scheme of the electrospark erosion experiment for the fabrication of MoS_2_@ZnO nanoheterostructures.

**Figure 2 nanomaterials-11-00157-f002:**
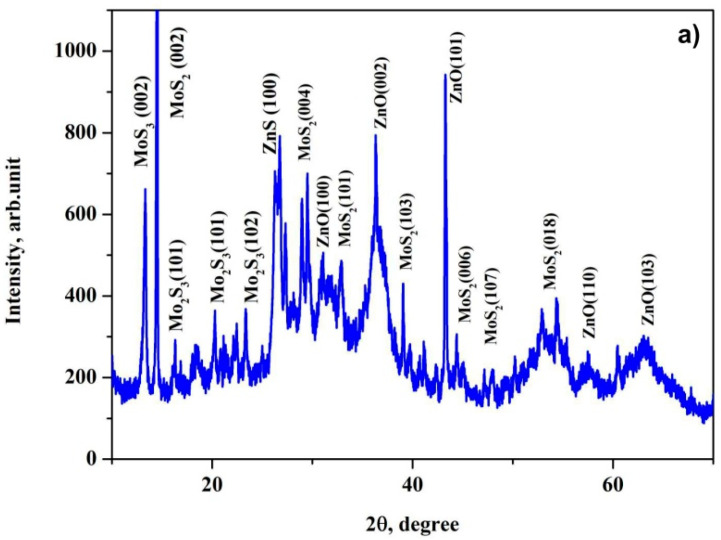
X-ray diffraction (XRD) patterns of the prepared MoS_2_@ZnO nanoheterostructures (**a**) and initial nanostructured MoS_2_ (**b**).

**Figure 3 nanomaterials-11-00157-f003:**
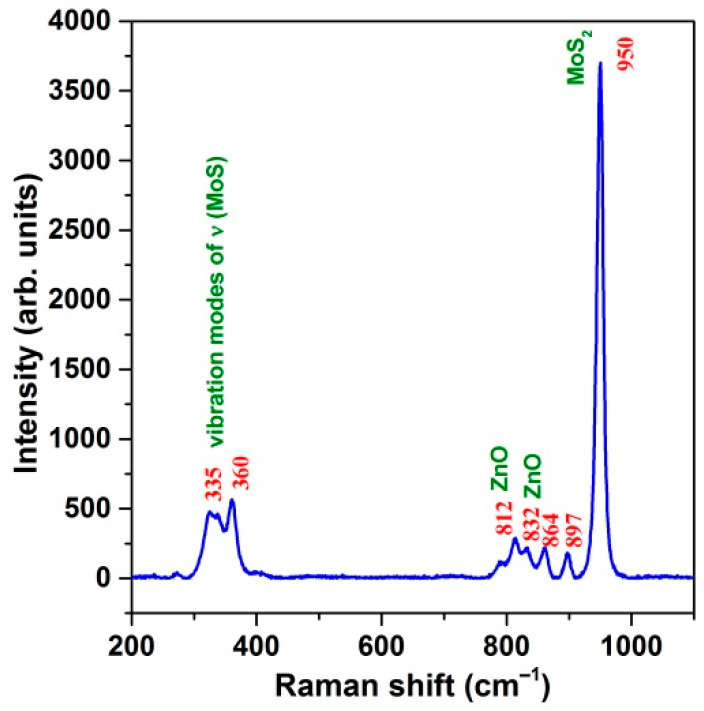
Raman spectrum of the prepared MoS_2_@ZnO nanoheterostructures.

**Figure 4 nanomaterials-11-00157-f004:**
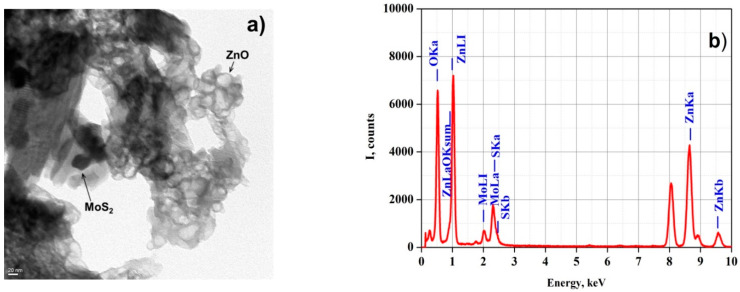
Transmission electron microscopy (TEM) image (magnification of 500,000×) (**a**) and energy-dispersive X-ray (EDS) spectrum (**b**) of the prepared MoS_2_@ZnO nanoheterostructures.

**Figure 5 nanomaterials-11-00157-f005:**
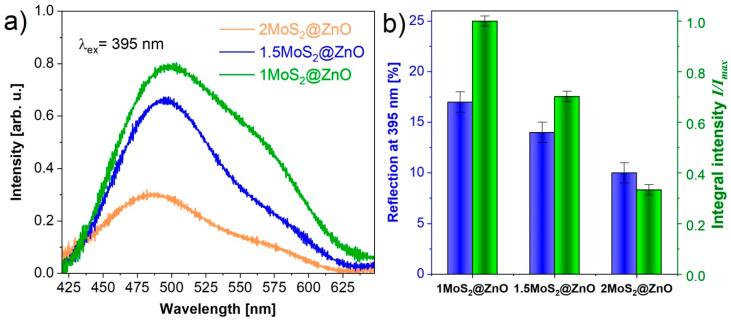
Photoluminescence (PL) spectra (**a**) and reflection chart (**b**) of the prepared MoS_2_@ZnO nanoheterostructures.

**Figure 6 nanomaterials-11-00157-f006:**
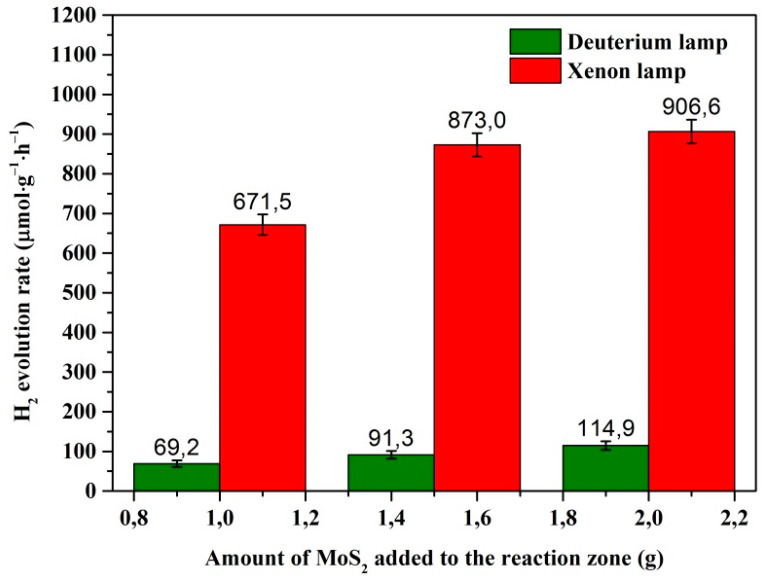
Rate of H_2_ production by MoS_2_–ZnO nanoheterostructures. Measurement conditions: 0.06 g sample, 100 mL aqueous solution of 0.1 M Na_2_S with 0.05 M Na_2_SO_3_, and light sources of 150 W Xe and 30 deuterium lamp.

## Data Availability

Data available in a publicly accessible repository.

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
