# Peer review of "MoS2@ZnO Nanoheterostructures Prepared by Electrospark Erosion for Photocatalytic Applications"

_nanomaterials, 2021, doi:10.3390/nano11010157_

Round 1
Reviewer 1 Report
The manuscript by An Vladimir et al. reports on MoS2/ZnO nano-heterostructures prepared by electrospark erosion for photocatalytic applications.
Authors demonstrate the successful preparation and synthesis of MoS2/ZnO nano-heterostructures which are additionally characterized by the XRD measurements, TEM analysis, Raman scattering measurements and PL experiments.
Overall this paper is well organized and conveys a clear message. I believe it can be published in Nanomaterials, after minor revision.
Below are my comments on the manuscript. I do hope that these comments will be helpful to the authors to improve the quality of their manuscript:
1) The TEM image of the MoS2/ZnO nanoheterostructures shows that the synthesized material represents two different types of nanoscale overlapping particles. Their size is in the order of tens nanometers. Intuitively, the Raman scattering spectra are changing for different positions on the nano-heterostructure. What is a diameter of laser spot in Raman scattering measurements?
Is the Raman spectrum presented in Fig. 3 measured for particular positions on the heterestructure? Is the Raman spectrum repetitive at different points of the heterostructure?
2) In lines 260-264 Authors write: “The luminescence is attributed to excess electrons in the conduction band… The result obtained suggests that the recombination rate of photogenerated carriers is significantly reduced and that the charge transfer between the interface of ZnO and MoS2 is evidently facilitated.”
How the PL spectrum changes as a function of increasing power excitation density? Cold Authors provide additional experiments and discuss the effect of increasing laser power excitation on the recombination of photo-generated carriers?

Author Response
Dear Reviewer,
We are pleased to get that our manuscript has received a positive review. Thank you for reviewing our manuscript and pointing out the significant comments. We think we have complied with your comments in our revised manuscript. We hope our manuscript can be accepted this time.
Responses to your comments:
1) The TEM image of the MoS2/ZnO nanoheterostructures shows that the synthesized material represents two different types of nanoscale overlapping particles. Their size is in the order of tens nanometers. Intuitively, the Raman scattering spectra are changing for different positions on the nano-heterostructure. What is a diameter of laser spot in Raman scattering measurements?
Is the Raman spectrum presented in Fig. 3 measured for particular positions on the heterestructure? Is the Raman spectrum repetitive at different points of the heterostructure?
Response: Many thanks for your valuable comment! The diameter of laser spot in Raman scattering measurements was 3 μm. This information was added to the Methods and Materials section. The Raman spectrum was repetitive at different points of the heterostructure and represents an integral image. The number of investigated areas was varied from 3 to 5.
2) In lines 260-264 Authors write: “The luminescence is attributed to excess electrons in the conduction band… The result obtained suggests that the recombination rate of photogenerated carriers is significantly reduced and that the charge transfer between the interface of ZnO and MoS2 is evidently facilitated.” How the PL spectrum changes as a function of increasing power excitation density? Cold Authors provide additional experiments and discuss the effect of increasing laser power excitation on the recombination of photo-generated carriers?
Response:
«The luminescence is attributed to excess electrons in the conduction band… it is suggestion related to Ref. [47] when we have changed the electron beam density. We have modified it in the revised version of the manuscript, lines 269 and 272.
How the PL spectrum changes as a function of increasing power excitation density?
The studies of photoluminescence upon excitation by different excitation sources showed no difference in the emission spectra.
Cold Authors provide additional experiments and discuss the effect of increasing laser power excitation on the recombination of photo-generated carriers?
Thank you so much for your question. Actually it will be quite interesting to conduct investigations with different types of excitation sources. We think that next step will focus on the research with different e-beam excitation density. In present paper we have demonstrated for scientific community the possibility of MoS2@ZnO nanoheterostructures synthesis by electrospark erosion technique. Also the characterization of these nanomaterials were investigated.

Reviewer 2 Report
In this paper by An and colleagues, the authors observed how MoS2@ZnO structures can be used for photocatalysis. The results are interesting and fit a vigorously explored area of research, but there are several issues that must be addressed first before the manuscript can be published. For now, major revisions are required. Upon introduction of the changes, I am more than happy to re-evaluate it.
1) The state of the art is well-described, but unfortunately, the "in this work" section is too brief. Please extend it for the readers to enable fast familiarization with the content of the article. What is more, in this part, the novelty factor should be clearly defined. Please be specific about what is new as compared with the previous contributions by other scientists in this area.
2) Experimental - sources of chemicals and materials should be reported along with the corresponding purity values and other crucial parameters. For instance, at present, one cannot know what kind of zinc granules were used in this study, which makes the results irreproducible by others. Moreover, what was the electrode separation distance in the MoS2@ZnO synthesis, and what was the voltage/current?
3) More details regarding characterization parameters should be given e.g. for Raman spectroscopy - acquisition time, number of integrations, objective, number of investigated areas.
4) The article formatting should be improved - remove the empty spaces, unify the font size among all the plots, etc.
5) Please provide an explanation of why XRD patterns are so noisy.
6) Raman features should have corresponding assignments in Fig. 3.
7) "Morever, a special role in the of The extreme non-equibrium conditions under which the MoS2@ZnO nanoheterostructures were prepared during electrospark erosion in the medium of hydrogen peroxide can be the origin of some of the observed anomalies in their Raman spectrum" (Lines 206-209). There is something wrong with this sentence.
8) Please redraw the EDX spectrum from Fig. 4 using raw data. Currently, it is not consistent with the formatting and very hard to read.
9) Please comment on the inhomogeneity of the material (Fig. 4a). Is it composed of distinct islands made of ZnO and MoS2? EDX mapping is recommended to investigate it.
10) Some of the legends (particularly from Fig. 5a) are unclear. Please correct them.
11) It would be interesting to evaluate the performance for HER of just MoS2 and ZnO separately to discern their individual contributions.
11) The key flaw of this study is that there is no error analysis. Not a single plot contains error bars, which puts doubt if the result interpretation is justified. Please conduct more experiments, do the statistics, and modify the descriptions accordingly.
Author Response
Dear Reviewer,
We are pleased to get that our manuscript has received a positive review. Thank you for reviewing our manuscript and pointing out the significant comments. We think we have complied with your comments in our revised manuscriptr. We hope our manuscript can be accepted this time.
Responses to the Reviewer’s comments:
1) The state of the art is well-described, but unfortunately, the "in this work" section is too brief. Please extend it for the readers to enable fast familiarization with the content of the article. What is more, in this part, the novelty factor should be clearly defined. Please be specific about what is new as compared with the previous contributions by other scientists in this area.
Response: Thanks a lot for this valuable comment. Of course, you are right that the «in this work» section should be enlarged according to your advice. We thoroughly revised the entire Introduction section and have made the following corrections:
The introduction section was completed with the information on what is the main feature of our method compared with other scientists' method. A distinctive feature of the developed method of heterostructure formation (covered in article) consists in the use of simple elements (Mo, S, Zn) to prepare metal oxide and sulfide. In addition, as opposite to autoclave synthesis the electrospark erosion was applied to create hydrothermal conditions of heterostructure formation.
2) Experimental - sources of chemicals and materials should be reported along with the corresponding purity values and other crucial parameters. For instance, at present, one cannot know what kind of zinc granules were used in this study, which makes the results irreproducible by others. Moreover, what was the electrode separation distance in the MoS2@ZnO synthesis, and what was the voltage/current?
Response: Thanks a lot for this valuable comment. Authors provided the additional information about chemicals and materials (molybdenum wire, sulfur, zinc granules, sodium salts, hydrogen peroxide solution); namely, the purity, the manufacturers. MoS2@ZnO synthesis was clarified by the description of experimental condition (electrode separation distance of 10 cm, the voltage of 500 V, the current of 150 A).
3) More details regarding characterization parameters should be given e.g. for Raman spectroscopy - acquisition time, number of integrations, objective, number of investigated areas.
Response: Thank you very much for this comment! The more detailed parameters was additionally given for Raman spectroscopy in the Materials and Methods section.
4) The article formatting should be improved - remove the empty spaces, unify the font size among all the plots, etc.
Response: Thanks a lot for this comment. The article style was carefully checked. The plot sizes were adjusted so that to get the plots with the same font size. The style of figure captions was corrected. Also some other changes were made according to Information for Authors.
5) Please provide an explanation of why XRD patterns are so noisy.
Response: Thanks a lot for this comment. The XRD patterns are noisy because the synthesized heterostructures are very small and amorphized which relates to the extreme conditions of electrospark erosion method.
6) Raman features should have corresponding assignments in Fig. 3.
Response: Thanks a lot for this comment. The Raman features were assigned in Fig.3. Figure 3 as replaced in the manuscript.
7) "Morever, a special role in the of The extreme non-equibrium conditions under which the MoS2@ZnO nanoheterostructures were prepared during electrospark erosion in the medium of hydrogen peroxide can be the origin of some of the observed anomalies in their Raman spectrum" (Lines 206-209). There is something wrong with this sentence.
Response: We are grateful to the reviewer for this comment! Of course there is an error of typing. We have revised this sentence and replaced it in the manuscript:
“Moreover, a special role of the extreme non-equibrium conditions under which the MoS2@ZnO nanoheterostructures were prepared during electrospark erosion in the medium of hydrogen peroxide can be the origin of some of the observed anomalies in their Raman spectrum.”
8) Please redraw the EDX spectrum from Fig. 4 using raw data. Currently, it is not consistent with the formatting and very hard to read.
Response: The EDX spectrum was redrawn and replaced in the manuscript.
9) Please comment on the inhomogeneity of the material (Fig. 4a). Is it composed of distinct islands made of ZnO and MoS2? EDX mapping is recommended to investigate it.
Response: The inhomogeneity of the material is connected to the conditions of the electrospark erosion, but, at the same time, they provide a possibility for MoS2 nanoparticles to adhere to newly formed zinc oxide particles.
10) Some of the legends (particularly from Fig. 5a) are unclear. Please correct them.
Response: The legend from Fig.5a was corrected and replaced in the manuscript.
11) It would be interesting to evaluate the performance for HER of just MoS2 and ZnO separately to discern their individual contributions.
Response: Thank you for this comment. Yes we plan to thoroughly investigate the performance for HER of just MoS2 and ZnO separately because now we have not enough time during the revision procedure (5 days).
11) The key flaw of this study is that there is no error analysis. Not a single plot contains error bars, which puts doubt if the result interpretation is justified. Please conduct more experiments, do the statistics, and modify the descriptions accordingly.
Response: Many thanks for your valuable comment. We will conduct another set of experiments in order to do more detailed statistics (very short of the revision (5 days) does not allow to conduct additional investigations).

Reviewer 3 Report
The authors report the realization of an heterostructure formed by MoS2 nanoparticles and ZnO by means of Hydrothermal synthesis. The obtained samples were examined using transmission electron microscopy and XRD analysis, Raman Photoluminescence spectroscopy.
It’s an interesting paper showing promising application of the heterostructures but the results must be better presented and commented. Revision are needed before any publication may be considered.
1- Nanostructured MoS2 was synthesized by a self-propagating high-temperature and then added to the solution containing ZnO. What is the exact nature of this nanostructures MoS2 powder. Is it layered bulk 2D MoS2 in the 2H semicondutive phase or metallic 1T phase? Is it a sort of amorphous 3D material? Can the author report its structural characterization by TEM, XRD and Raman?
2- Raman spectra are presented at low spectral resolution. The author must show the higher resolution Raman spectra in the 350-450 cm-1 and 750-1000 cm-1 bands.
3- Raman spectra of 2D layered MoS2 shows main peaks at 380 cm-1 and 410 cm-1 corresponding to in-plane and out-of-plane vibrational modes (see Lee et al, ACS Nano 2010, 4, 5, 2695–2700 https://pubs.acs.org/doi/10.1021/nn1003937). Here the band at 335-360 cm-1 is attributed to MoS2. What is the reason of this shift in the Raman spectra? Is due to a different phase of MoS2? Is it the effect of Oxygen defect? Is it due to the electronic coupling to ZnO?
4- The XRD pattern refer to the formed heterostructure MoS2-ZnO. In the text they cite but do not report the results of XRD of isolated ZnO and MoS2 . It would be interesting compare the XRD spectra of isolated material with that of the heterostructure.
5- In their introduction the authors mention several recent papers demonstrating the formation of type II heterostructure MoS2-ZnO with efficient electron transfer from MoS2 to ZnO and hole transfer in the opposite direction. However here they also report in the XRD comment on the formation of interfacial layer between the two materials. What is the band level alignment of such heterostructure?
6- The PL data show wide emission in 425 -625 band. This emission is far from that of layered MoS2 (650-700nm) and that of ZnO (<400nm). It is also worth notice that the interlayer gap in a Type II heterostructure must be lower (in energy ) than that of each of the two isolated materials. For such a type II heterostructure we should expect interlayer emission at wavelength larger than 700nm. What is the origin of the measured emission? The author must better comment their PL data and propose a band level alignment scheme supporting their data and explanations.
7- The figure 5 data report labels (likely referring to the relative content) which are not well explained neither in the main text nor in the figure caption.
Author Response
Dear Reviewer,
We are pleased to get that our manuscript has received a positive review. Thank you for reviewing our manuscript and pointing out the significant comments. We think we have complied with your comments in our revised manuscript. We hope our manuscript can be accepted this time.
Responses to your comments:
1- Nanostructured MoS2 was synthesized by a self-propagating high-temperature and then added to the solution containing ZnO. What is the exact nature of this nanostructures MoS2 powder. Is it layered bulk 2D MoS2 in the 2H semicondutive phase or metallic 1T phase? Is it a sort of amorphous 3D material? Can the author report its structural characterization by TEM, XRD and Raman?
Response: Thank you for your comment. Molybdenum disulfide produced by self-propagating high-temperature synthesis (SHS) is 2H-MoS2. The article is completed by the X-ray diffraction pattern of the SHS product from molybdenum and sulfur. Raman spectroscopy has not been performed yet, but the authors plan to carry out experiments on high-resolution Raman spectroscopy of all initial reagents as well as the obtained nanoheterostructures.
2- Raman spectra are presented at low spectral resolution. The author must show the higher resolution Raman spectra in the 350-450 cm and 750-1000 cm bands.
Response: We plan to conduct a special investigation including the higher resolution Raman spectra in the areas you mentioned. We are persuaded that we can find additional information of the Raman shifts specific for synthesized heterostructures.
3- Raman spectra of 2D layered MoS2 shows main peaks at 380 cm and 410 cm corresponding to in-plane and out-of plane vibrational modes (see Lee et al, ACS Nano 2010, 4, 5, 2695–2700 https://pubs.acs.org/doi/10.1021/nn1003937). Here the band at 335-360 cm is attributed to MoS . What is the reason of this shift in the Raman spectra? Is due to a different phase of MoS2? Is it the effect of Oxygen defect? Is it due to the electronic coupling to ZnO?
Response: Thank you for this valuable comment. The shift in the Raman spectra is because of oxygen defects. This results in the formation of molybdenum oxysulfides in the surface and near-surface layers of particles. It is mentioned in the manuscript.
4- The XRD pattern refer to the formed heterostructure MoS2- ZnO. In the text they cite but do not report the results of XRD of isolated ZnO and MoS2. It would be interesting compare the XRD spectra of isolated material with that of the heterostructure.
Response: Thanks a lot for this comment. There is no X-ray diffraction pattern of zinc oxide. It is due to zinc oxide was formed at electrospark erosion of zinc granules in a solution of hydrogen peroxide containing the nanostructured molybdenum disulfide. So zinc oxide was formed as a part of the MoS2@ZnO nanoheterostructure.
5- In their introduction the authors mention several recent papers demonstrating the formation of type II heterostructure MoS2-ZnO with efficient electron transfer from MoS2 to ZnO and hole transfer in the opposite direction. However here they also report in the XRD comment on the formation of interfacial layer between the two materials. What is the band level alignment of such heterostructure?
Response: Thanks a lot for this comment. As mentioned in Response 4, the extreme fabrications conditions, including the plasma formation, can result in the formation of molybdenum oxysulfides in the surface and near-surface layers of particles. It allows us to predict the band alignment of the MoS2/ZnO-interface can be changed from type II or to type III or misaligned one.
6- The PL data show wide emission in 425 -625 band. This emission is far from that of layered MoS2 (650-700nm) and that of ZnO (<400nm). It is also worth notice that the interlayer gap in a Type II heterostructure must be lower (in energy) than that of each of the two isolated materials. For such a type II heterostructure we should expect interlayer emission at wavelength larger than 700nm. What is the origin of the measured emission? The author must better comment their PL data and propose a band level alignment scheme supporting their data and explanations.
Response: We considered the possible origin emission of ZnO in the manuscript using suitable references which support our experimental results. In Ref. [43. Lv, J.; Li, C.; Chai, Z. Defect luminescence and its mediated physical properties in ZnO. Journal of Luminescence 2019, 208, 225-237, doi: 10.1016/j.jlumin.2018.12.050] the emission of ZnO nanocrystals considered in detail. The authors have shown that the emission in the spectral region of 494 nm (2.51 eV) is due to VO. Emission in the spectral region of 450 nm can be attributed to an interstitial zinc defect. The general nature of luminescence is due to intercellular interstitial zinc and oxygen. This reasoning is well illustrated in Ref. [43]. We have cited this article in our work.
The electronic energy levels of native imperfections in ZnO well described in [44. Schmidt-Mende, L.; MacManus-Driscoll, J. L. ZnO–nanostructures, defects, and devices. Materials Today 2007, 10(5), 40-48, DOI: 10.1016/S1369-7021(07)70078-0]. The band level alignment scheme is presented in Fig. below. Fig. shows that there are a number of defect states within the bandgap of ZnO. In this case which defect dominates in native, undoped ZnO is still a matter of great controversy.
Energy levels of native defects in ZnO. The donor defects are Z i**, Z i*, Z ix, V0**, V0*, V0
and the acceptor defects are Vzn’’, Vzn’ (Adapted from Kröger) [44]
There are a number of intrinsic defects with different ionization energies. The Kröger Vink notation uses: i = interstitial site, Zn = zinc, O = oxygen, and V = vacancy. The terms indicate the atomic sites, and superscripted terms indicate charges, where a dot indicates positive charge, a prime indicates negative charge, and a cross indicates zero charge, with the charges in proportion to the number of symbols.
According to the Ref. [44] we did not make the same diagrams in the manuscript and only cite it.
7- The figure 5 data report labels (likely referring to the relative content) which are not well explained neither in the main text nor in the figure caption.
Response: Fixed. The Fig. 5 was changed. We added required information in the Figure.
Figure 5. PL spectrum and reflection chart of the prepared xMoS2@ZnO heterostructures (x=1, 1.5, 2 g.)

Round 2
Reviewer 2 Report
Thank you for your response. However, some comments were not handled appropriately:
8) Again, please use the raw data, paste it into plotting software, and produce a plot like Fig. 3. At present, Fig. 4 is not readable.
11) Ensuring reproducibility of the study is a must. If necessary, request an extension from the editorial office to provide an error analysis. Without this, the article is not rigorous enough to be accepted for publication.
Author Response
Dear Reviewer,
Thank you for reviewing our manuscript and pointing out the significant comments. We think we have complied with your comments in our revised manuscript. We hope our manuscript can be accepted this time.
Responses to the Reviewer’s comments:
8) Again, please use the raw data, paste it into plotting software, and produce a plot like Fig. 3. At present, Fig. 4 is not readable.
Response: Thanks a lot for this valuable comment. The additional days given by the Editor for the revising work in order to eliminate the shortcomings shown in your comments allowed us to treat carefully the raw EDX data and to redraw Figure 4b. We incorporated redrawn Figure 4b into the manuscript.
11) Ensuring reproducibility of the study is a must. If necessary, request an extension from the editorial office to provide an error analysis. Without this, the article is not rigorous enough to be accepted for publication.
Response: Thanks a lot for this valuable comment. The Authors undertook an error analysis in order to demonstrate reproducibility of the study. We conducted this analysis for some performed experiments (except the spectral measurements where this is normally connected to standard accuracies of the well-known spectral methods) – for the reflection measurements and HER experiments. As a result of this analysis, we redrew Figures 5b and 6 where we input the error bars made according to our analytical estimations and then reincorporate the figures into the revised manuscript. The following sentences were added to the manuscript:
1) Line 261: “The reflection measurement accuracy was ±1%.”
2) Lines 289-293: “The accuracy of the hydrogen evolution rate measurements is within standard deviation. According to the error analysis for the HER measurements, the standard deviations depend on the amount of MoS2 added to the reaction zone and were in the ranges of 8.3-10.7 mmol×g-1×h-1 and 25.9-30.1 mmol×g-1×h-1 for the experiments with the use of the deuterium and xenon lamp, respectively”.

Reviewer 3 Report
In the revised version of their paper the authors have addressed all the points raised in my previous review.
The paper deserves to be published as it is in the present form
Author Response
Dear Reviewer,
We are pleased to get that our manuscript has received a positive review. Thank you for reviewing our manuscript and pointing out the significant comments. We think we have complied with your comments in our revised manuscript. We hope our manuscript can be accepted this time.
Response to your comments:
In the revised version of their paper the authors have addressed all the points raised in my previous review.
The paper deserves to be published as it is in the present form
Response: Thanks a lot for your positive decision!

Round 3
Reviewer 2 Report
Thank you. The article can now be accepted for publication.